# Maternal Amino Acid Status in Severe Preeclampsia: A Cross-Sectional Study

**DOI:** 10.3390/nu14051019

**Published:** 2022-02-28

**Authors:** Natasya Prameswari, Rima Irwinda, Noroyono Wibowo, Yudianto Budi Saroyo

**Affiliations:** 1Department of Obstetrics and Gynaecology, Faculty of Medicine, Universitas Indonesia/Cipto-Mangunkusumo Hospital, Jakarta 10340, Indonesia; 2Maternal Fetal Division, Department of Obstetrics and Gynaecology, Faculty of Medicine, Universitas Indonesia/Cipto-Mangunkusumo Hospital, Jakarta 10340, Indonesia; rima.irwinda@gmail.com (R.I.); wibowonoroyono@yahoo.com (N.W.); yudibs@gmail.com (Y.B.S.)

**Keywords:** preeclampsia, amino acid, oxidative stress

## Abstract

Introduction: Preeclampsia has been one of the leading causes of maternal death in Indonesia. It is postulated that its relationship with oxidative stress may be the underlying pathology of the disease. Nutrients and amino acids have been suggested as a scavenger for oxygen-free radicals. No previous study regarding the amino acid status in preeclampsia has been conducted in women in Indonesia. Methods: This was a cross-sectional study of a total of 64 pregnant women, 30 with normal pregnancy and 34 with severe preeclampsia. Data were obtained in Cipto Mangunkusumo National Referral Hospital in Jakarta from July to December 2020. Maternal blood samples were taken during or soon after delivery. Amino acid levels were analyzed using liquid chromatography-tandem mass spectrometry (LC-MS/MS). Bivariate analysis was then performed. Results: We identified 19 different levels of amino acids in this study. Four amino acids that were elevated in the preeclampsia group were phenylalanine, serine, glycine, and glutamate. Serine (331.55 vs. 287.43; *p* = 0.03), glycine (183.3 vs. 234.35, *p* = 0.03), and glutamate levels (102.23 vs. 160.70, *p* = 0.000) were higher in preeclamptic patients. While in the essential amino acids group, phenylalanine levels (71.5 vs. 85.5, *p* = 0.023) were higher, and methionine levels (16.3 vs. 12.9, *p* = 0.022) were lower in preeclamptic patients. Conclusions: These findings suggest that severe preeclampsia had differences in concentration of some amino acids compared to normal pregnancy. Glutamate and methionine were associated with preeclampsia. Furthermore, a more detailed study regarding amino acids in the pathomechanism of preeclampsia is suggested.

## 1. Introduction

Preeclampsia is one of the most common complications in pregnancy, as it is one of the Great Obstetrics Syndromes. Pregnancy is complicated by oxidative stress from maternal decidua and placental villus. Insufficient blood flow in the placenta can give rise to a hypoxic environment that causes ischemia reperfusion injury, marked by increasing free radicals in cells [1]. The severity of the damage depends on the cellular defense to oxidative stress [2]. Furthermore, preeclampsia is based on a complex placental pathology that is related to poor trophoblast differentiation or placental immaturity [3]. Insufficient decidualization may lead to placental hypoxia and, moreover, endothelial and vascular dysfunction manifested as preeclampsia [4].

Preeclampsia has been one of the leading causes of death in the world, as 10% of pregnancies are complicated by preeclampsia. In Indonesia, hypertension in pregnancy is reported as the second leading cause of maternal death after hemorrhage [5,6]. In Cipto Mangunkusumo General Hospital Jakarta, around 24.3% of pregnant women who undergo delivery have hypertension in pregnancy every year [7].

Nutrients, such as amino acids, have a role in immunomodulatory properties and interact through common biochemical pathways. Other than being used in energy metabolism reactions and synthesis of proteins, amino acids are important for the synthesis of antioxidants, nitric oxide, histamine, and hydrogen peroxide [8]. A previous study by Bahado-Singh et al. showed an increased concentration of branch-chained amino acids (BCAA) in preeclamptic patients. Dysfunctions in amino acid transport in the placenta can occur in preeclampsia and intrauterine growth restrictions. An imbalance of BCAA levels in some studies has been found to be associated with insulin resistance and preeclampsia [9].

A previous study in Jakarta, Indonesia, reported that almost all pregnant women in Jakarta had low nutrient intake in their first trimester, below their recommended dietary allowance. Most of the subjects (80.8%) also had a low protein intake of less than 56 g per day [10]. Nutrients may modify certain inflammatory responses and also modulate pro-inflammatory cytokine production and action. It has also been suggested that some amino acids can directly scavenge oxygen free radicals [11]. It has been suggested that high concentrated branched-chain amino acids, such as leucine, isoleucine, and valine, induce productions of reactive oxygen species and pro-inflammatory status in peripheral blood mononuclear cells [12].

Previous studies showed higher amino acids related to preeclampsia [13,14]. However, there are no studies regarding amino acid status in preeclampsia and normal pregnant women in Indonesia. This study aims to evaluate the profile of the maternal serum amino acids in preeclampsia and normal pregnancy in order to be used as a basis for the prediction of preeclampsia or as supplementation to reduce the risk of preeclampsia.

## 2. Materials and Methods

### 2.1. Study Participants

This was a cross-sectional study of a total of 64 pregnant women. Out of 64 pregnant women, 30 were normal pregnancies compared to the 34 patients with severe preeclampsia. Following delivery, all participants had their blood examined, as well as given the food frequency questionnaire. Data were obtained in Cipto Mangunkusumo National Referral Hospital in Jakarta from July to December 2020. This study was approved by the Ethical Committee for Research in Humans from the Faculty of Medicine, Universitas Indonesia (KET-799/UN2.F1/ETIK/PPM.00.02/2019). All of the participants have also given their informed consent prior to their inclusion in the study.

Preeclampsia with severe features as defined by ACOG 2013, and the recently updated ACOG practice bulletin on Gestational Hypertension and Preeclampsia [15], include the criteria of blood pressure ≥ 160/110 mmHg, thrombocytopenia, progressive renal insufficiency, new onset cerebral or visual disturbances, and pulmonary edema. Criteria for proteinuria is expected to be 300 mg or more in 24 h urine collection, protein dipstick of 2+, or protein/creatinine ratio of 0.3 mg/dL. In the absence of proteinuria, we included other criteria that match preeclampsia with severe features based on ACOG [15]. Exclusion criteria in this study include multiple pregnancy, intrauterine growth restriction, congenital anomalies, HIV-positive patients, preterm premature rupture of membrane, diabetes mellitus, infection, and autoimmune diseases.

### 2.2. Nutrient Intake Assessment

Nutrient intake from the subjects was evaluated using a semiquantitative food frequency questionnaire (FFQ) for a month. The FFQ was assessed by trained nutritionists using a food model and was converted to the Nutrisurvey program with the food database from Indonesia. The classification of the nutrient intake of protein was based on the recommended dietary allowances (RDA).

### 2.3. Sample Preparation and Amino Acid Measurements

Maternal blood samples were taken during or soon after delivery. Samples were collected using venous puncture into 5 mL tubes (Vacutainer; Becton-Dickinson). The serum was then taken to the laboratory and was directly separated out from the whole blood and frozen at −80 °C until all samples were analyzed. Amino acid levels were analyzed using liquid chromatography-tandem mass spectrometry (LC-MS/MS) 6460 Triple Quad with 1290 Infinity Binary Pump (Agilent Technologies^®^, USA) and converted into µM (µmol/L).

### 2.4. Outcome Measures

The data obtained for the characteristics of the subject include maternal age, education level, parity, gestational age, and body mass index.

The concentrations of maternal serum amino acids were then measured for both essential and non-essential amino acids. The essential amino acids examined were lysine, histidine, threonine, valine, methionine, isoleucine, leucine, phenylalanine, and arginine. Whereas the non-essential amino acids examined were aspartate, ornithine, serine, glycine, alanine, tyrosine, cystine, cysteine, and glutamate. All concentrations of the amino acid were expressed in µM (µmol/L).

### 2.5. Statistical Analysis

Data were analyzed using Statistical Package for Social Sciences (SPSS) version 25.0 (IBM, United States). The numeric data were firstly checked for normal distribution using the Kolmogorov–Smirnov test, then presented as mean ± SD if normally distributed and as median (min-max) if not normally distributed. Unpaired t-test or Mann–Whitney test were used to identify significances among groups.

In addition, receiver operating characteristics (ROC) curve analyses were used to determine cut-off values, sensitivity, and specificity of amino acid variables. To predict preeclampsia, we assessed the area under the curve (AUC). The cut-off values for parameters with high performances were determined at the highest value of sensitivity and specificity. Chi-square test was performed for bivariate analysis, followed by multivariate logistic regression for maternal age, gestational age, and pre-pregnancy BMI adjustment. Results corresponding to *p*-values < 0.05 were considered significant.

## 3. Results

### 3.1. Characteristics of Subjects

All of the samples were included in this study. Table 1 shows the characteristics of the subjects included. There were significant differences found between the groups in terms of maternal age, parity, mode of delivery, BMI, and fetal weight. The BMI values were categorized as underweight (BMI below 18.5), normal (BMI range 18.5 to 22.9), overweight, and obese (BMI above 23). Patients with preeclampsia had higher maternal age of 31.7 (7.3) years old (*p* < 0.05), multiparity (*p* < 0.05), higher rate of cesarean section (*p* < 0.05), fetal weight lower than control (*p* < 0.05), and most BMI were >23 kg/m^2^ (*p* < 0.05).

### 3.2. Nutrient Intake within Subjects

The daily maternal nutrient intake is shown in Table 2. The mean daily intake of energy and carbohydrate in normal patients were higher than in preeclampsia patients. The mean protein and fat intake of preeclampsia patients were higher compared to normal patients. In protein intake, we found that most subjects in both normal and preeclampsia groups had intakes that were below the Estimated Average Requirement and RDA recommendations of 1.1 g/kg body weight per day [16].

### 3.3. Amino Acids Levels in Normal Pregnancy and Preeclampsia

The levels of serum amino acids, both essential and non-essential, are shown in Table 3. The preeclampsia group was found to have higher levels of most essential and non-essential amino acids.

### 3.4. ROC Curve Analysis

Amino acids that had the AUC of ROC curves > 0.65 were considered as appropriate indicators for severe preeclampsia, and thus, included in further analyses. Essential and non-essential amino acids with AUC < 0.65 were not taken into account.

In predicting the risk of preeclampsia, glutamate and glycine had AUCs of 0.775 and 0.658, respectively, while methionine had an AUC of 0.672 (Figure 1). The glutamate value ≥ 109 had 73.5% sensitivity and 66.7% specificity, the glycine value of ≥187.5 had 61.8% sensitivity and 53.3% specificity, and the methionine value of ≤14.5 had 67.6% sensitivity and 63.3% specificity.

### 3.5. Bivariate and Multivariate Analysis of the Cut-Off Points

In the bivariate analysis, only glutamate and methionine cut-off values were found to be significantly associated with severe preeclampsia (*p* = 0.003 and *p* = 0.026). After adjusting for the maternal age, gestational age, and pre-pregnancy BMI, the high-risk levels of glutamate and methionine showed increased risks of severe preeclampsia with OR 5.89, 95%CI 1.85–18.76; and OR 3.75, 95%CI 1.24–11.3; respectively (Table 4).

## 4. Discussion

Our research was mostly dominated by overweight and obese participants. This finding was supported by a previous study by Ermamilia et al. [17] in Yogyakarta, Indonesia, where BMI ≥ 23 kg/m^2^ was a predictive factor of preeclampsia.

Higher levels of both non-essential (serine, glycine, and glutamate) and essential amino acids (phenylalanine) were detected in patients with preeclampsia compared to normal pregnancy. This finding correlates with previous findings by Liu G et al., which compared 18 amino acids in serum and cord blood samples of preeclamptic patients and normal pregnant patients [14]. The study found that glutamate and serine levels were higher in maternal blood samples of those with preeclampsia compared to normal ones. Higher nitrogen requirements of the placenta that correspond to the state of higher oxidative level in preeclampsia may give rise to the higher levels of glutamate and serine in the maternal blood [14].

Hypertension and the upregulated cycle of inducible nitric oxide (iNOS) are reported both clinically and experimentally. Glutamate induced and increased iNOS production, proteins, and also Ca2+. Therefore, an increase in glutamate may result in higher levels of iNOS in preeclamptic patients [18]. The higher levels of serine and glycine in preeclampsia were thought to be in connection with glycolysis defects in the placenta of preeclamptic patients. Glucose is the main source of nutrition in the placenta. Preeclampsia affects the glycolysis function, therefore causing failure in metabolism. The decreased glycolysis enzyme in the placenta causes low pyruvate and lactate productions, consequently producing less energy for the fetus [19,20].

Our study showed lower methionine levels in preeclamptic patients compared to normal pregnancy. Methionine is a sulfuric amino acid that is important in protein synthesis. Some suggested that methionine can act as an immune regulatory factor, like IL-1 or insulin-like growth factor I. Methionine can protect cells from oxidative damage, which can also play a role in preeclampsia [21,22,23,24]. Methionine is also a precursor of homocysteine. Unfortunately, in our study, both folate and B12 levels were not analyzed. Some studies linked low methionine levels to B12 deficiency as cofactor methionine synthase. Folate and B12 deficiency may affect methionine, S-adenosylmethionine (SAM), and higher homocysteine due to the ineffectiveness in extracting methionine from homocysteine [19]. Methionine has been suggested to provoke transsulfuration that leads to methionine catabolism and also re-methylation through homocysteine [25,26]. In our study, low levels of methionine can also be caused by failure of homocysteine conversion to methionine due to low B12 levels.

A previous study found that 74.8% of the first trimester pregnant women in Jakarta, Indonesia had low daily maternal folate intake but high maternal folate in serum [10]. Methylenetetrahydrofolate reductase (MTHFR) is a key enzyme in regulating folate balance and homocysteine conversion to methionine [27]. MTHFR and homocysteine levels are associated with preeclampsia, since previous studies have found the correlation between MTHFR polymorphism and preeclampsia [27,28]. In this study, homocysteine, vitamin B12, and MTHFR were not evaluated; therefore, future studies should evaluate the role of MTHFR enzymes.

Our findings suggest that severe preeclampsia had significant differences in the concentrations of some amino acids compared to normal pregnancy. Nevertheless, our study did not give a clear view of the causal relationship between amino acids and preeclampsia since our samples were limited to only maternal serum. The study by Liu et al. revealed that amino acid levels differ in maternal and cord blood of preeclampsia patients. It might show that the different levels of amino acids have an influence on fetal health related to fatty acid oxidation [14]. Preeclampsia is a complicated process that can result in abnormal changes in the metabolism of both fetal and maternal serum that can result in adverse consequences. Abnormalities in the amino acid pathways have not yet been fully understood until now.

Previous studies showed that fasting state is related to the concentration of amino acids [29]. In our study, all of the serum amino acid levels were taken without the patients fasting. Therefore, a more elaborate study that defines whether the serum is taken during fasting or postprandial state is suggested. We also suggest a more detailed study regarding the role of amino acids in the pathomechanism of preeclampsia. However, this is the first study in Indonesia that investigated the amino acid profile in maternal serum of preeclamptic patients.

## 5. Conclusions

Glutamate, serine, glycine, and phenylalanine amino acid levels were elevated in the maternal serum of preeclampsia patients. Glutamate increases the risk of preeclampsia by 5.5 times, whereas low methionine levels can increase the risk of preeclampsia by 3.6 times. We suggested a more detailed study regarding maternal intake, maternal serum, cord blood, placental amino acid levels, and their correlations with oxidative stress and inflammation in preeclampsia patients.

## Figures and Tables

**Figure 1 nutrients-14-01019-f001:**
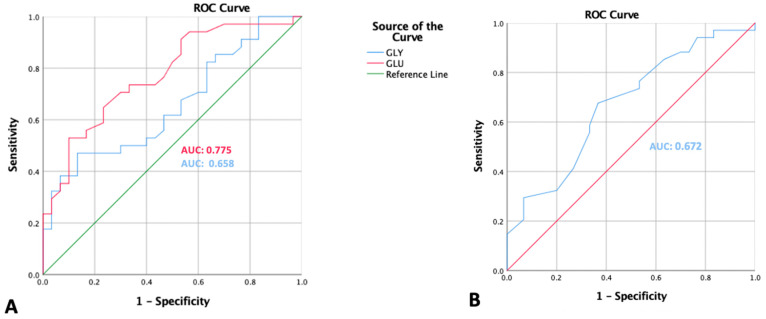
ROC curves of amino acids that were considered to be acceptable predictors of severe preeclampsia. (**A**) Glycine and glutamate. (**B**) Methionine.

**Table 1 nutrients-14-01019-t001:** Characteristics of subjects (*n* = 64).

Variables	Control (*n* = 30)	Preeclampsia (*n* = 34)	*p*
Maternal age (years)	26.9 ± 5.9	31.7 ± 7.3 *	0.01
Education level (*n*/%)			
Low education level	23 (76.7%)	31 (91.2%)	0.111
High education	7 (23.3%)	3 (8.8%)	
Gestational age at delivery (weeks)	38.7 (1.03)	35.56 (3.4)	0.06
Parity (*n*/%)			
Nuliparity	19 (70.4%)	8 (29.6%)	0.001
Multiparity	11 (29.7%)	26 (70.3%)	
Body mass index (*n*/%)			
Underweight	4 (13.3%)	2 (5.9%)	
Normal	11 (36.7)	7 (20.6)	0.053
Overweight and obese	15 (50%)	25 (73.5%)	
Mode of delivery			
Vaginal delivery	22 (73.3%)	9 (26.5%)	<0.005
Caesarean Section	8 (26.7%)	25 (73.5%)	
Fetal Weight (grams)	3091.67 ± 395.4	2386.91 ± 828.1	<0.005

Data presented as mean ± SD. * Significances to control (*p* < 0.05).

**Table 2 nutrients-14-01019-t002:** Maternal nutrient intake.

Daily Maternal Intake	Control	Preeclampsia	*p*
Energy (kcal)	1809.367 ± 707.19	1792.207 ± 759.59	0.869 *
Carbohydrate (g)	244.711 ± 80.81	219.100 ± 112.28	0.167 *
Protein (g)	60.867 ± 30.81	130.539 ± 334.37	0.148 *
Fat (g)	70.093 ± 37.17	76.582 ± 31.12	0.162 *

Data presented as mean ± SD. * Mann–Whitney Test.

**Table 3 nutrients-14-01019-t003:** Amino acids levels in normal pregnancy and preeclampsia.

Variables	Control (*N* = 30)	Preeclampsia (*N* = 34)	*p*
Essential Amino Acids			
Arginine	159.33 (44.5)	172.70 (39.9)	0.885
Histidine	66.6 (18.8)	74.02 (18.7)	0.330
Isoleucine	37.2 (12.7)	40.4 (14.6)	0.651
Leucine	84.03 (30.9)	91.05 (34.8)	0.726
Lycine	139.03 (62–307)	150.44 (71–336)	0.258
Methionine	16.3 (5.8)	12.9 (5.7)	0.022 *
Phenylalanine	71.5 (20.5)	85.5 (26.8)	0.023 *
Threonine	166.1 (55.66)	193.588 (72.23)	0.096
Valine	111.56 (38.1)	111.8 (36.0)	0.591
Non-essential amino acids			
Aspartate	45.3 (15.50)	47.5 (15.5)	0.461
Serine	287.43 (85–398)	331.55 (174–620)	0.03 *
Glycine	183.3 (59.2)	234.35 (213.00)	0.03 *
Cysteine	4.36 (1.00–7.00)	6.44 (1.00–45.00)	0.788
Alanine	413.233 (196–822)	417.64 (146.98)	0.824
Glutamate	102.23 (29–183)	160.70 (36–397)	0.000 *
Proline	114.1 (57.7)	120.88 (56.1)	0.483
Tyrosine	43.7 (12.1)	44.7 (10.09)	0.931
Ornitine	40.4 (15.15)	47.1 (24.3)	0.121

Data presented as mean ± SD and median. * Significance to control (*p* < 0.05).

**Table 4 nutrients-14-01019-t004:** Amino acid levels and risk for preeclampsia.

	Preeclampsia *n *(%)**	Control *n *(%)**	*p* Bivariate	OR (95%CI)	Adjusted OR (95%CI) *
Glutamate					
High risk (≥109)	25 (73.5)	10 (33.3)	0.003	5.55 (1.89–16.28)	5.89 (1.85–18.76)
Low risk (<109)	9 (26.5)	20 (66.7)		1.0	1.0
Glycine					
High risk (≥187.5)	21 (61.8)	14 (46.7)	0.33	31.84 (0.68–5.00)	
Low risk (<187.5)	13 (38.2)	16 (53.3)		1.0	
Methionine					
High risk (≤14.5)	34 (67.6)	11 (36.7)	0.026	3.61 (1.28–10.1)	3.75 (1.24–11.3)
Low risk (>14.5)	11 (32.4)	19 (63.3)		1.0	1.0

* Adjusted for covariates: maternal age and maternal pre-pregnancy BMI. 1.0: Reference Value.

## Data Availability

The data used in this study can be requested from the corresponding author upon reasonable request.

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
