# Peer review of "Maternal Amino Acid Status in Severe Preeclampsia: A Cross-Sectional Study"

_nutrients, 2022, doi:10.3390/nu14051019_

Round 1

Reviewer 1 Report

Dear Authors!

 The article is devoted to an interesting topic, made at a high technical level, however, there are a few remarks. The text of the article is well written, does not duplicate the data of tables and figures.

1.Why was this particular set of amino acids chosen, and not the full panel of all available ones was used?

  1. It is necessary to indicate the gestational age at the time of delivery in the groups of patients with PE and control for a correct assessment of the data obtained, since the gestational age at which pregnant women were delivered can affect the level of amino acid composition of the blood.
  2. Can amino acid levels depend on gestational age? If not, can you provide a relevant reference (correct data)? Therefore, the control group should include pregnant women with comparable (with the PE group) gestational age.
  3. The results of the study do not provide data on the frequency of operative delivery, prenatal rupture of amniotic fluid, etc., which, from a hypothetical point of view, can also correlate with the level of amino acids.
  4. The article does not provide equations for models built on the basis of the use of logistic regression, which does not allow obstetrician-gynecologists who are ignorant of regression analysis to use and test the information content of the developed models in practice.

 Similar article has already been published. Kan, N.E., Lomova, N.A., Amiraslanov, E.Y., Chagovets, V.V., Tyutyunnik, V.L., Khachatryan, Z.V., Starodubtseva, N.L., Kitsilovskaya, N.A., Frankevich, V.E. Specific features of a metabolomic profile in preeclampsia (2019) Akusherstvo i Ginekologiya (Russian Federation), 2019 (11), pp. 82-88. DOI: 10.18565/aig.2019.11.82-88.

Reviewer 2 Report

The authors presented an article on a not new but current problem - preeclampsia and the role of the amino acid status of the mother during pregnancy and the development of its complications (preeclampsia). The novelty, apparently, is that an analysis was carried out in a single state - in Indonesia.
The patient sample is small but sufficient for statistical processing and analysis.
In my opinion, the discovered patterns confirm the data that researchers presented in similar works earlier.
The results of this brief article may be of interest to those skilled in the art and may be the subject of a more detailed study of the mechanisms for the development of preeclampsia in various pathological conditions permissible in endothelial dysfunction.
At the same time, the data itself do not carry any new information, but only confirm that the change in the level of methionine, phenylalanine, glycine, cysteine and glutamate can be associated with a severe course of pregnancy and the development of preeclampsia.
Also strange is the choice of amino acids for analysis - in the article you should explain why these amino acids are chosen for analysis. The authors do not make assumptions that have led to such changes in amino acid composition - perhaps this is just a change in eating behavior, or it is endothelial dysfunction with the initiation of lipid peroxidation and inflammation processes. It would be interesting to see the results of the analysis of the correlation of amino acid composition with the activity of lipid peroxidation and inflammation processes - it is they that are proposed by the authors as the next steps in the development of the topic of the article.

Reviewer 3 Report

In this study the authors found that phenylalanine, serine, glycine, and glutamate levels were increased in Preeclampsia. In addition, phenylalanine levels were higher and methionine levels were lower in preeclamptic patients.

The study is interesting but the manuscript needs substantial improvements:

  • Lines 30-35: The authors should stress the complexity of PE pathology since it has been also reported that PE is also characterised by trophoblast immaturity (PMID: 32529396) and vascular dysfunction (PMID: 34831277). 
  • Line 184: Authors should underline the fact that oxidative stress can induce severe endothelial dysfunction (PMID: 34153425; PMID: 33123312), which plays a key role in preeclampsia.
  • Table 1: The BMI thresholds used for Normal, undeweight, Overweight and obese classification must be reported. Moreover, Foetal weight must be reported in order to show possible Foetal Growth Restriction (FGR).
  • Figure 1: the AUC values should be added 
  • A thorough grammar and syntax review of English is recommended

Round 2

Reviewer 1 Report

Dear Authors!

Many corrections and additions were made in the article, comments were taken into account. However, when re-examining the article, I have some remarks.

The criteria for preeclampsia presented in the article are presented in accordance with ACOG 2013, but they were revised in 2017. 

  These criteria should be given, as well as the level of proteinuria.  It is also clarified how the authors confirmed the specificity of the antibody anti-b2GP1, given that this antibody was obtained in the laboratory, and not the antibody of the official manufacturer.

Reviewer 3 Report

the manuscript has been significantly improved

Author Response

Thank you for your response and comment.